# Germination Performances of 14 Wildflowers Screened for Shaping Urban Landscapes in Mountain Areas

Matteo Caser *, Sonia Demasi, Eric Mozzanini, Paola Maria Chiavazza and Valentina Scariot

Department of Agricultural, Forest and Food Sciences, University of Torino, Largo Paolo Braccini 2, 10095 Grugliasco, TO, Italy; sonia.demasi@unito.it (S.D.); eric.mozzanini@unito.it (E.M.); paolamaria.chiavazza@unito.it (P.M.C.); valentina.scariot@unito.it (V.S.)
* Correspondence: matteo.caser@unito.it; Tel.: +39-011-670-8935

**Abstract:** Despite the high biodiversity in the Mediterranean region, the use of wildflowers from mountain areas in urban landscaping projects is hampered by the limited information on their seed germination performances. This research evaluated germination traits of 14 native herbaceous wildflower species from northern west Italian alpine and subalpine areas. Seed germination in Petri dishes at 25 °C was performed, applying two different photoperiod conditions (light/dark at 0/24 h or 12/12 h). A high rate and rapid germination are key features for seed and seedling nursery production; thus, the main germination indices were evaluated: the final germination percentage, the index of germination relative to light, the time of first germination, the time to reach 50% of germinated seeds, the germination period, and the mean germination time. Overall, *Bellis perennis* L., *Leucanthemum vulgare* Lam., and *Taraxacum officinale* Weber, from mesophilic mountainous hilly grasslands, and *Dianthus carthusianorum* L. and *Lavandula angustifolia* Mill. from higher altitudes, turned out to be interesting. Particular attention could be paid to *D. carthusianorum* for germination synchrony in both growth conditions, high speed, and short germination period (8.5 and 16.2 days in the dark and in the light, respectively).

**Keywords:** alpine plant species; final germination percentage; germinability; germination period; native flowering species; native plants biodiversity; photoperiod; relative light germination percentage; wildflower species



## 1. Introduction

The European Environment Agency [1] states that the 'quality of life in cities depends on the existence of sufficient attractive urban green areas for people and wildlife to thrive'. Recent research indicated that the biodiversity in green areas, both within and outside cities, should be increased by using native wildflowers [2–4]. These are defined as combinations of annual or perennial herbaceous wild species suitable for sowing in low maintenance areas [5–8]. The multifunctional role of wildflowers is already well-known, as they help promote biodiversity and restore habitat, requiring low-input management [3,6,9–11].

Native plants are generally well adapted to the local pedo-climatic conditions, since they have evolved over hundreds of years to thrive in the soil and in the climate of their original area and are expected to better adapt when used in urban landscaping projects [12]. Thus, herbaceous native wildflowers can rapidly colonize poor soils, reducing erosion risk, providing natural pest control and an attractive view [13]. Recent studies reported the successful use of wildflower seed mixtures in urban areas such as urban parks, roundabouts, and green roofs, but also in buffer stripes and walls for urban vertical gardens [2,3,10,14,15]. Studies on habitat restoration are also available, using wildflowers for roadside revegetation and mine rehabilitation and for improving runoff quality [10,16,17]. However, most of the literature investigated Mediterranean species such as *Calendula arvensis* L., *Centaurea aplolepa* Moretti, *Trifolium campestre* Schreb., *Nigella damascena* L., and *Silene conica* L. [2–4,18,19]. Until now, only a few data have been available for alpine species [16].

In the alpine and subalpine zones, the large species and microhabitat diversity provide an ideal context to assess habitat-related regenerative strategies, depending on habitat provenance, main microhabitat, chorotype, and adaptation to local environmental conditions [20]. Urban and alpine areas possess stress similarities such as the presence of soils with low fertility [21–23]. Thus, alpine and subalpine wildflowers can be a possible choice to ensure plants survival in urban and stressed areas by exploiting their ability to grow in stressed habitats. For these reasons, a deeper investigation of alpine and subalpine species is required to understand their potential.

Wildflowers, however, may also have other lesser known features in order to introduce new insights into the trade of urban horticulture and propagating plants. In fact, since ancient times, wild plants have widely been used in traditional cultures for different purposes, such as food [24]. Today, the renewed interest in wild edible plants, supported by the knowledge of the healthy phytochemicals they contain, makes them definable as new functional foods, which may be used also in edible landscaping projects [25–29].

Italy has less than 3% of the European surface but supports the highest number of both animal and plant species within the European Union, as well as the highest rate of endemism [30]. Even though most native plants are well-known from a botanical point of view, little information on their germination aptitude is available [7]. The lack of specific studies on rare or endemic species is due to several limitations such as their restricted geographic distribution, the difficulty in their identification, and the absence of economic interest. The deficiency of this type of knowledge is the reason why nurseries specialized in alpine native plants propagation and the related market have not been properly developed yet. The seeds used for the reproduction of flowering and ornamental plants must possess certain requirements that meet the needs of the nursery industry. Among these, there is purity, both genetic, i.e., the guarantee of belonging to a single variety or species, and technological, i.e., not being mixed with various impurities (earth, crushed stone), and vitality. The knowledge of the environmental conditions that promote seed germination also plays a crucial role in professional lawn mixes creation. Among these, light has been seen to have varied effects on germinating seeds of different plants [31]. In some species, such as *Alopecurus myosuroides* Huds. [32], seeds need light to germinate (photoblastic seeds), whereas in others, such as *Glaucium flavum* Crantz. [33], the germination is hindered by light (non-photoblastic seeds).

This research aims to unveil the potential of herbaceous flowering plants from alpine and subalpine areas as sources of biodiversity for urban landscaping in mountain areas. For this purpose, the germination performances, under the presence or absence of light, was evaluated in 14 native wildflower species of the western Alps.

## 2. Materials and Methods

### 2.1. Seed Collection

Mature seeds of 14 native species (Figure 1) were collected in the northern west Italian alpine and subalpine areas (Table 1) during spring–summer 2017. They were cleaned from inert matter and other small debris, closed in paper bags, and kept at 4 °C without pre-dried treatment until the germination test, performed in the laboratory of the Department of Agricultural, Forest, and Food Sciences (DISAFA) of the University of Turin (Italy), during 2017 and 2018. Wild species were selected to explore all altitudinal belts in the studied area, including plain, hills, montane and alpine belts, and for their peculiar ornamental traits.

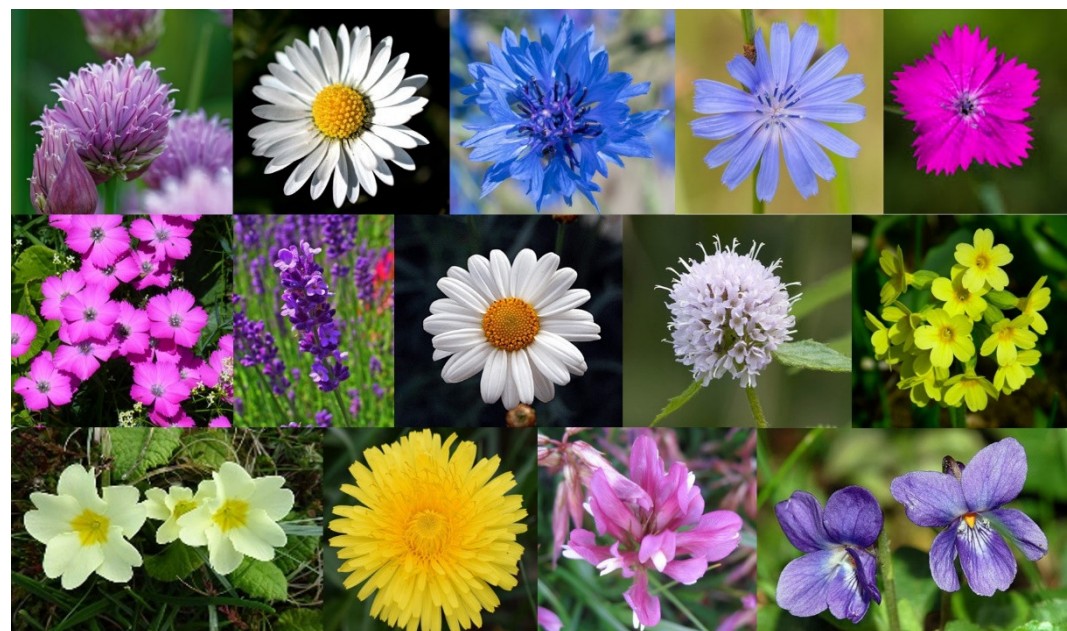

**Figure 1.** Detail of the flower of the fourteen native species selected for this study. From left to right, first line: *Allium schoenoprasum* L., *Bellis perennis* L., *Centaurea cyanus* L., *Cichorium intybus* L., *Dianthus carthusianorum* L.; second line: *Dianthus pavonius* Tauesch, *Lavandula angustifolia* Mill., *Leucanthemum vulgare* Lam., *Mentha aquatica* L., *Primula veris* L.; third line: *Primula vulgaris* Hudson, *Taraxacum officinale* Weber, *Trifolium alpinum* L., *Viola odorata* L.

**Table 1.** Taxonomy, vegetation community and geographical site references of the 14 studied alpine and sub-alpine wildflowers.

| Plant Species | Family | Site of Collection | Geographic References | Altitude (m a.s.l.) |
|---|---|---|---|---|
| *Allium schoenoprasum* L. | Amaryllidaceae | Castelmagno (CN) | Long 7.122 Lat 44.390 | 2293 |
| *Bellis perennis* L. | Asteraceae | Grugliasco (TO) | Long 7.592 Lat 45.065 | 285 |
| *Centaurea cyanus* L. | Asteraceae | Ivrea (TO) | Long 7.896 Lat 45.475 | 240 |
| *Cichorium intybus* L. | Asteraceae | Chivasso (TO) | Long 7.843 Lat 45.192 | 189 |
| *Dianthus carthusianorum* L. | Caryophyllaceae | Balme (TO) | Long 7.219 Lat 45.301 | 1435 |
| *Dianthus pavonius* Tauesch | Caryophyllaceae | Castelmagno (CN) | Long 7.122 Lat 44.391 | 2393 |
| *Lavandula angustifolia* Mill. | Lamiaceae | Grugliasco (TO) | Long 7.592 Lat 45.065 | 287 |
| *Leucanthemum vulgare* Lam. | Asteraceae | Grugliasco (TO) | Long 7.593 Lat 45.065 | 285 |
| *Mentha aquatica* L. | Lamiaceae | Caselette (TO) | Long 7.485 Lat 45.120 | 364 |
| *Primula veris* L. | Primulaceae | Cesana Torinese (TO) | Long 6.802 Lat 44.968 | 1395 |
| *Primula vulgaris* Hudson | Primulaceae | Cesana Torinese (TO) | Long 7.379 Lat 45.145 | 774 |
| *Taraxacum officinale* Weber | Asteraceae | Grugliasco (TO) | Long 7.593 Lat 45.064 | 285 |
| *Trifolium alpinum* L. | Fabaceae | Castelmagno (CN) | Long 7.122 Lat 44.390 | 2385 |
| *Viola odorata* L. | Violaceae | Grugliasco (TO) | Long 7.591 Lat 45.065 | 287 |

### 2.2. Thousand Seed Weight (TSW)

Seeds were sorted into eight replicates of 100 seeds per each species. For each replicate, weight was recorded in grams to three decimal places by using a scale (Kern EW 220-3NM, Kern&Sohn GmbH, Balingen, Germany), and the mean weight was determined from eight values [34]. The mean weight of 100 seeds was then used to calculate the weight of 1000 seeds (TSW). Thus, the number of seeds per gram was measured.

### 2.3. Germination Test

Based on the International Rules for Seed Testing Association (IRSTA) [34], seed germination at 25 °C was performed applying two different photoperiod conditions (light/dark at 0/24 h or 12/12 h; 25–30 µmol m$^{-1}$ s$^{-2}$ under cool, fluorescent white lamps). One hundred seeds were used for each species per treatment (10 repetitions by 10 seeds). Seeds were placed to germinate in 9 cm lidded Petri dishes containing Whatman

No. 1 filter paper soaked in 5 mL of distilled water. Seeds were considered germinated when seedling structures (emergence of the radicle at the peduncle-end of the seed) were visible. Non-germinated seeds were classified as dead or vane.

Based on IRSTA [34], data were collected daily until germination values became unchanged. Six germination indices were calculated: final germination percentage (FGP), relative light germination percentage (RLGP), first germination time (FGT), halftime of germination ($T_{50}$), germination period (GPD), and mean time of germination (MTG). The formulas used to calculate the indices, explanations and references are given in Table 2.

**Table 2.** Germination indices and formulas used to evaluate alpine and subalpine wildflowers, with explanations and references.

| Germination Index | Formula | Explanation | Reference |
|---|---|---|---|
| Final germination percentage (FGP) | $FGP = 100 * GN/SN$ | GN = total number of germinated seeds; SN = total number of seeds tested. | [35] |
| Relative light germination percentage (RLGP) | $RLGP = Pl/(Pd + Pl)$ | RLGP is an expression of the light requirement for germination. Pl = percentage germination in light; Pd = percentage germination in shade. | [36] |
| First germination time (FGT) | | Number of days from the beginning of the experiment to first germination. | [37] |
| Half time of germination ($T_{50}$) | | Number of days from the beginning of the experiment to the count reached 50% of the final germination. | [38] |
| Germination period (GPD) | | Number of days from the beginning of the experiment to the maximum number of seeds germinated. | [39] |
| Mean germination time (MGT) | $MGT = \Sigma(NS * DAS)/GN$ | NS = number of germinated seeds; DAS = days after sowing; GN = total number of germinated seeds. Calculation is based on the daily count of normal seedling until the final date of the germination test. | [40] |

RLGP is expressed in values ranging from 0 (germination in dark only) to 1 (germination in light only). Therefore, species can be classified as light-dependent ($1.0 > RLGP > 0.6$), light-inhibited ($0 < RLGP < 0.4$), or intermediate ($0.4 < RLGP < 0.6$) [36].

Species germination performances were assessed also according to germination potential (GP), germination pattern (GPa), and germination response (GRe) as indicated by Wu et al. [31]. GP was defined as either high (FGP > 75%), moderate (FGP 74–20%), or low (FGP < 19%). The GPa was defined as either synchronous, if 90% of seeds germinated within 15 days after FGT, or asynchronous in all other cases. The GRe was defined as fast (MGT < 10 days), moderate (10 days < MGT < 20 days), or slow (MGT > 20 days).

*2.4. Tetrazolium Assay*

The tetrazolium (TZ) assay was performed as described by Wharton [41] with slight modifications to screen for seed viability. Seeds of the species showing low final germination percentage (FGP $\leq$ 15%), were first divided into slices and soaked with distilled water. After 24 h, slices were soaked in a 1% solution of 2,3,5-triphenyl tetrazolium chloride (Sigma-Aldrich S.r.l., Milan, Italy) and incubated in darkness at 35 °C for 24 h. TZ precipitates to red-colored 2,3,5-triphenyl formazan by the activity of dehydrogenases present in the live cells. As a result, viable seeds containing live cells stain red and nonviable or dead seeds remain unstained. Thus, the viability of seeds can be interpreted by the staining pattern and the color intensity. Data were expressed as percentage of viability. Heat-killed seeds (incubated at 100 °C for 1 h) were used as a negative control.

*2.5. Statistical Analysis*

An arcsin transformation was performed on all percentage incidence data before statistical analysis in order to improve the homogeneity of the variance (Levene test; $p < 0.05$). All the analyzed data were checked for the normality of variance by using a

Shapiro–Wilks test ($p > 0.05$). For all the analyzed parameters not respecting the ANOVA assumptions, mean differences among species were computed using a non-parametric Kruskal–Wallis test ($p < 0.05$) by step-wise comparison. Mean differences among light treatments were computed by Mann–Whitney U-test. Statistical analyses were performed using SPSS Version 25.0 (IBM SPSS Statistic, Armonk, NY, USA). When the germination of each species was equal to or above 50% of the total number of seeds in the germination trials, the cumulative values of seed germination were plotted against time. Pearson correlation among the studied traits was conducted by using PAST 4.0 software (Natural History Museum, University of Oslo, Norway).

## 3. Results and Discussion

Germination is an irreversible process and must be timed to occur when the environment is favorable for subsequent seedling establishment [20]. Germination timing is controlled both by environmental and morphological cues such as seed weight. In the present study, species' weight differed greatly among species (Table 3). The magnitude of seed weight and number of seeds per gram ranged from 0.023 g and 43,478 seeds for *L. vulgare* to 5.710 g and 175 seeds for *T. alpinum*, with a mean of 1.198 g per 1000 seeds.

**Table 3.** Thousand seed weight (TSW) and number of seeds per gram of the studied species. Mean data of TSW are presented ± standard deviation.

| Plant Species | TSW (g) | Seeds g$^{-1}$ (n.) |
|---|---|---|
| *Allium schoenoprasum* | 0.968 ± 0.011 | 1033 |
| *Bellis perennis* | 0.615 ± 0.030 | 1626 |
| *Centaurea cyanus* | 3.440 ± 0.030 | 290 |
| *Cichorium intybus* | 0.117 ± 0.001 | 8547 |
| *Dianthus carthusianorum* | 0.086 ± 0.001 | 11,628 |
| *Dianthus pavonius* | 0.534 ± 0.042 | 1873 |
| *Lavandula angustifolia* | 0.096 ± 0.001 | 10,417 |
| *Leucanthemum vulgare* | 0.023 ± 0.001 | 43,478 |
| *Mentha aquatica* | 0.122 ± 0.001 | 8197 |
| *Primula veris* | 0.734 ± 0.023 | 1362 |
| *Primula vulgaris* | 0.966 ± 0.018 | 1035 |
| *Taraxacum officinale* | 0.061 ± 0.001 | 16,393 |
| *Trifolium alpinum* | 5.710 ± 0.202 | 175 |
| *Viola odorata* | 3.308 ± 0.078 | 302 |

Several authors highlighted the role of light availability in seed germination performances of wild species [4,42–44]. In the present study, significant differences for all the studied parameters related to seed germination were observed in both light conditions. In *C. cyanus*, *P. veris*, *P. vulgaris*, and *V. odorata*, no seeds germinated in both tested photoperiods (Table 4). These species were therefore not considered in subsequent statistical analyses.

Among the other species, in the darkness condition, FGP ranged from 0% for *M. aquatica* and *A. schoenoprasum* to 90% for *L. vulgare* and *D. carthusianorum*, while, under 12 h of light, between 1% for *M. aquatica* to 96% in *L. vulgare*. Within species, significant effects of light conditions were observed only for *A. schoenoprasum*, *T. officinale*, and *L. angustifolia* seeds, which performed better under 12 h of light. With respect to the germination potential (GP) of these species (Table 4), only *A. schoenoprasum* showed different behavior on the basis of light availability, with low germination potential in darkness and moderate under 12 h of light. Overall, two species have high potential (*L. vulgare* and *D. carthusianorum*), four moderate (*B. perennis*, *C. intybus*, *T. officinale*, and *L. angustifolia*), and the others low. FGP is the most important characteristic of the seed to consider in order to adopt the most effective germination protocol. Seed germination of wild species can be naturally low and variable, and some seed ecological traits can determine obstacles [16,45]. Our results, with no specific pattern, are congruent with Bu et al. [46] which assumed that

inferences about inter-species variation in seed size and their effects on germination cannot be generalized. In fact, with an increase in seed mass, germination percentage may increase, decrease, or remain the same [31]. Furthermore, in the alpine environment, the large species and microhabitats diversity have resulted in a variety of germination responses, which makes difficult to define common germination strategy [47].

**Table 4.** Effect of dark (0/24 h) and photoperiod of 12 h of light (12/12 h) on final germination percentage (FGP, %) of the studied species. Germination potential (GP) was described as high (FGP > 75%), moderate (FGP 74–20%), or low (FGP < 19%). Relative light germination percentage (RLGP) was calculated on the FGP and species were classified as light dependent (LD; 1.0 > RLGP > 0.6), light inhibited (LI; 0 < RLGP < 0.4), or intermediate (LInt; 0.4 < RLGP < 0.6).

| Species | FGP (%) | | | GP | | RLGP (Classification) |
|---|---|---|---|---|---|---|
| | 0/24 | 12/12 | $p$ | 0/24 | 12/12 | |
| Allium schoenoprasum | 0 f | 23.0 e | *** | Low | Moderate | 1.00 (LD) |
| Bellis perennis | 66.0 b | 70.0 bc | ns | Moderate | Moderate | 0.51 (LInt) |
| Centaurea cyanus | 0 f | 0 g | ns | Low | Low | - |
| Cichorium intybus | 41.0 c | 47.0 d | ns | Moderate | Moderate | 0.53 (LInt) |
| Dianthus carthusianorum | 90.0 a | 87.0 ab | ns | High | High | 0.49 (LInt) |
| Dianthus pavonius | 9.0 e | 8.0 f | ns | Low | Low | 0.47 (LInt) |
| Lavandula angustifolia | 26.3 d | 67.6 c | *** | Moderate | Moderate | 0.72 (LD) |
| Leucanthemum vulgare | 90.0 a | 96.0 a | ns | High | High | 0.52 (LInt) |
| Mentha aquatica | 0 f | 1.0 g | ns | Low | Low | - |
| Primula veris | 0 f | 0 g | ns | Low | Low | - |
| Primula vulgaris | 0 f | 0 g | ns | Low | Low | - |
| Taraxacum officinale | 63.0 b | 77.0 bc | * | Moderate | Moderate | 0.55 (LInt) |
| Trifolium alpinum | 15.0 de | 8.0 f | ns | Low | Low | 0.35 (LI) |
| Viola odorata | 0 f | 0 g | ns | Low | Low | - |
| $p$ | *** | *** | | | | |

Means followed by the same letter in a column do not differ significantly, according to the Kruskal–Wallis test (*** = $p < 0.001$). Mean differences among light treatments were computed by the Mann–Whitney U-test (ns = non-significant, * = $p < 0.05$, *** = $p < 0.001$).

Regarding the light requirement for germination (RLGP) (Table 4), two species were light-dependent (*A. schoenoprasum* and *L. angustifolia*), one was light-inhibited (*T. alpinum*) and the remaining were intermediate. The mean RLGP value for the studied species was 0.57. Even if the light is indicated as an important factor for plant germination, in the present study, its positive role in the germination process of wild species was confirmed only for few species. This dependence was well-rendered in literature for very small seeds such as *Crepis bursifolia* L. [4,36,43]. Wu et al. [31] reported that light-dependent germination may also be related to physiological dormancy, a trait that can be considered as a germination strategy in response to habitat heterogeneity.

A TZ assay was performed to check the seed viability of species with low FGP. Results in Table 5 show high values of viability for all the tested seeds, ranging between 94% in *T. alpinum* and 98% in *V. odorata* and *M. aquatica*. A common alpine and non-alpine germination strategy is difficult to define; in fact, many plants may require, or not, deep physiological dormancy [48–50], different light conditions, and temperatures [51–53].

**Table 5.** Seed viability (%) test in tetrazolium assay.

| Plant Species | Seed Viability (%) |
|---|---|
| Centaurea cyanus | 96 |
| Dianthus pavonius | 95 |
| Mentha aquatica | 98 |
| Primula veris | 95 |
| Primula vulgaris | 97 |
| Trifolium alpinum | 94 |
| Viola odorata | 98 |

Thus, germination failure may be due to dormancy processes that characterize many wild species [54]. Consequently, knowing how to remove dormancy represents a pivotal factor for their cultivation. Our results confirmed dormancy processes for those species as previously detected by other authors. *Centaurea cyanus* showed a low FGP also in a germination trial performed at 5 and 20 °C [55]. Similarly, *V. odorata* seeds already showed typical hard physical dormancy [56]. In addition, previous studies have been suggesting that the seeds of Violaceae species have a seed coat with a mucilaginous inner layer that contains inhibitors, interfering with seed germination [57,58]. Conversely, cold-treatment and low temperature are required for *P. vulgaris* to reach germination [59]. *Allium sphaerocephalon* reached 88% of final germination, but with the use of in vitro technologies and gibberellic pre-treatment application [60].

Significant differences among species were also observed for time to first germination (FGT) and mean time of germination (MGT) traits (Table 6).

**Table 6.** Effect of dark (0/24 h) and photoperiod of 12 h of light (12/12 h) on first germination time (FGT, days) and mean time of germination (MGT, days) of the studied species. Germination pattern (GPa) was described as either synchronous (S) or asynchronous (A). Germination response (GRe) was described as fast (F), moderate (M), or slow (S).

| Species | FGT (Days) | | | GPa | | MGT (Days) | | | GRe | |
|---|---|---|---|---|---|---|---|---|---|---|
| | 0/24 | 12/12 | *p* | 0/24 | 12/12 | 0/24 | 12/12 | *p* | 0/24 | 12/12 |
| *Allium schoenoprasum* | - | 17.8 a | - | S | S | - | 21.9 a | - | - | S |
| *Bellis perennis* | 10.0 b | 4.8 bc | *** | A | A | 18.5 b | 11.3 b | *** | M | M |
| *Centaurea cyanus* | - | - | - | - | - | - | - | - | - | - |
| *Cichorium intybus* | 4.0 d | 3.8 cd | ns | S | S | 5.8 d | 5.7 c | ns | F | F |
| *Dianthus carthusianorum* | 4.0 d | 6.4 b | * | S | S | 5.9 d | 9.2 b | * | F | F |
| *Dianthus pavonius* | 9.6 b | 22.5 a | * | S | A | 11.7 c | 25.7 a | ** | M | S |
| *Lavandula angustifolia* | 28.6 a | 16.4 a | *** | A | A | 35.2 a | 25.8 a | *** | S | S |
| *Leucanthemum vulgare* | 6.2 c | 3.0 d | *** | S | S | 10.8 c | 4.3 d | *** | M | F |
| *Mentha aquatica* | - | - | - | - | - | - | - | - | - | - |
| *Primula veris* | - | - | - | - | - | - | - | - | - | - |
| *Primula vulgaris* | - | - | - | - | - | - | - | - | - | - |
| *Taraxacum officinale* | 4.2 d | 3.0 d | ns | S | S | 11.3 c | 9.5 b | ns | M | F |
| *Trifolium alpinum* | 4.5 cd | 5.2 cd | ns | S | S | 6.3 d | 5.7 cd | ns | F | F |
| *Viola odorata* | - | - | - | - | - | - | - | - | - | - |
| *p* | *** | *** | | | | *** | *** | | | |

Means followed by the same letter do not differ significantly, according to the Kruskal–Wallis test (*** = $p < 0.001$). Mean differences among light treatments were computed by the Mann–Whitney U-test (ns = non-significant, * = $p < 0.05$, ** = $p < 0.01$, *** = $p < 0.001$).

Regarding FGT, in darkness, the values ranged from 4 to circa 30 days. The fastest species to germinate were *C. intybus*, *D. carthusianorum*, *T. officinale*, and *T. alpinum* (4.0, 4.0, 4.2, and 4.5 days, respectively). Conversely, the slowest was *L. angustifolia* with 28.6 days. Under 12 h of light, data ranged from 3 to more than 22 days. *Cichorium intybus*, *T. officinale*, and *T. alpinum* were again the quickest to germinate with 3.8, 3.0, and 5.2 days, respectively, together with *L. vulgare* (3.0). In contrast, *A. schoenoprasum*, *D. pavonius*, and *L. angustifolia* were the slowest (17.8, 22.5, and 16.4 days, respectively). Within species, significant light-dependent differences were observed. *Bellis perennis*, *L. vulgare*, and *L. angustifolia* germinated faster in the light and *D. carthusianorum* and *D. pavonius* in the dark. As per the GPa, only the seed germination of *B. perennis* and *L. angustifolia* was asynchronous in both light conditions (Table 6). The remaining seven species were synchronous with the exception for *D. pavonius*, which showed different behavior, synchronous in the dark and asynchronous in the light. Asynchronous germination strategy is commonly adopted to enhance the probability of successful survival under varying environmental conditions [61]. This is generally observed for small-seeded species that may obtain a competitive advantage in time and space [31]. However, in the present study, this happened only in few small-seeded species (i.e., *B. perennis*, *L. angustifolia*, and *D. pavonius*). Regarding MGT, data ranged from 5.8 and 5.7 days for *C. intybus* to 35.2 and

25.8 days for *L. angustifolia* in the dark condition and 12 h of light, respectively. The dark condition induced a significant increase of MGT in *B. perennis*, *L. angustifolia* and *L. vulgare* and a reduction in *D. carthusianorum* and *D. pavonius* In addition, regarding GRe, we found that *L. angustifolia* resulted the only species with a slow germination response in both light conditions. In contrast, *A. schoenoprasum* and *D. pavonius* were slower under 12 h of light. In this last experimental condition, *T. officinale* and *L. vulgare* seed germination moved from moderate to fast. Hence, these germination strategies may be an adaptation to variable environments. Once there is a suitable environment, rapid germination could result in the seedlings of these species taking possession of empty space in a short time [31]. However, some factors in light chamber studies are different from natural conditions, such as moisture that is constant and optimal, filter paper that is sterile, and germination inhibitors that are probably more effectively washed out [62].

The germination time courses of the different genotypes in the different light conditions, whose FGP exceeded 50% [3] (*B. perennis*, *T. officinale*, *L. vulgare*, and *D. carthusianorum* in dark condition and also *L. angustifolia* in 12 h of light) are illustrated in Figure 2. Generally, for these species, wider lag phases (i.e., the interval from the beginning of the experiment to the onset of germination) were observed in the dark condition (Figure 2A) than with the application of 12 h of light (Figure 2B), with the exception for *D. carthusianorum*. This happened especially for *B. perennis* and *L. vulgare* seeds. These results are in agreement with those of Letchamo and Gosselin [63] and Keller [64], in which longer light period promotes faster and uniform germination for *T. officinale* and *L. vulgare*.

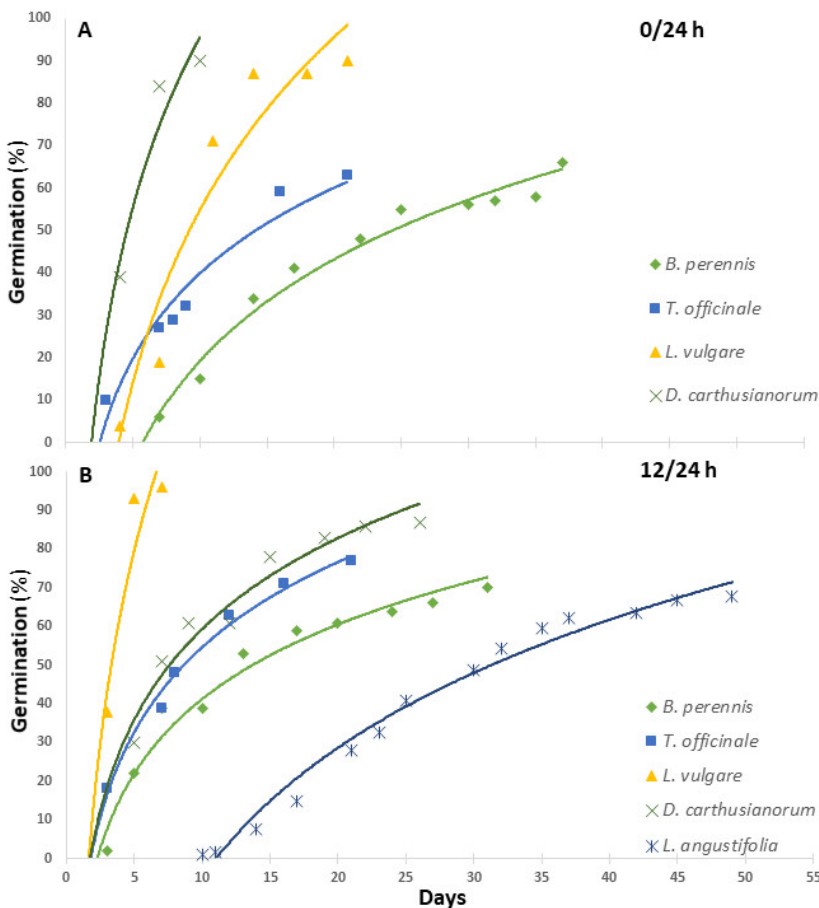

**Figure 2.** Germination time courses of the different species in (**A**) dark (0/24 h); and in a photoperiod (**B**) of 12 h of light (12/12 h); cumulative data on seed germination of each species were represented when germination was equal to or above 50%.

Germination clearly plays a key role in the successful establishment of wildflower communities in urban landscaping. It is also important for creating a dense vegetation cover, which controls weeds, and provides the ecological benefits of a rich flowering vegetation [3]. As a consequence, some other parameters, such as rapid and uniform germination, are essential when selecting wildflower species as putative candidates for the establishment of a vegetation cover [65]. Therefore, the half time of germination and the germination period are also important features in the establishment of seed mixtures because species with a good germination percentage, a rapid onset of germination, and low $T_{50}$ could be very useful and suitable for creating species-rich communities. In the present study, significant differences among species were observed in all these traits (Table 7).

**Table 7.** Effect of dark (0/24 h) and photoperiod of 12 h of light (12/12 h) on halftime of germination ($T_{50}$, days) and germination period (GPD, days) of the studied species.

| Species | $T_{50}$ (Days) | | | GPD (Days) | | |
|---|---|---|---|---|---|---|
| | 0/24 | 12/12 | $p$ | 0/24 | 12/12 | $p$ |
| *Allium schoenoprasum* | - | 20.8 a | - | - | 26.6 b | - |
| *Bellis perennis* | 16.8 b | 7.4 bc | *** | 31.8 b | 23.1 bc | * |
| *Centaurea cyanus* | - | - | - | - | - | - |
| *Cichorium intybus* | 4.6 d | 4.7 d | ns | 9.2 def | 9.3 d | ns |
| *Dianthus carthusianorum* | 5.8 d | 7.8 bc | ns | 8.5 ef | 16.2 c | ** |
| *Dianthus pavonius* | 9.7 c | 24.0 a | ** | 13.8 de | 30.5 ab | ns |
| *Lavandula angustifolia* | 35.0 a | 25 a | *** | 39.8 a | 36.6 a | * |
| *Leucanthemum vulgare* | 11.0 c | 5.6 cd | *** | 15.1 d | 5.6 d | *** |
| *Mentha aquatica* | - | - | - | - | - | - |
| *Primula veris* | - | - | - | - | - | - |
| *Primula vulgaris* | - | - | - | - | - | - |
| *Taraxacum officinale* | 11.0 c | 9.1 b | ns | 17.5 c | 15.5 c | ns |
| *Trifolium alpinum* | 5.0 d | 5.2 cd | ns | 6.0 f | 6.4 d | * |
| *Viola odorata* | - | - | - | - | - | - |
| $p$ | *** | *** | | *** | *** | |

Means followed by the same letter do not differ significantly, according to the Kruskal–Wallis test (*** = $p < 0.001$). Mean differences among light treatments were computed by the Mann–Whitney U-test (ns = non-significant, * = $p < 0.05$, ** = $p < 0.01$, *** = $p < 0.001$).

Concerning $T_{50}$, data ranged from 4.6 and 4.7 days for *C. intybus* to 35 and 25 days for *L. angustifolia* in dark condition and 12 h of light, respectively. Meanwhile, for GPD, data ranged from 6.0 days for *T. alpinum* to 39.8 days for *L. angustifolia* in dark, and between 5.6 days for *L. vulgare* to 36.6 days for *L. angustifolia* in 12 h of light. Within species, the dark condition induced a significant increase of both traits in *B. perennis, L. angustifolia*and *L. vulgare* and reduced GPD in *D. carthusianorum*. It has to be considered that species that germinate in few days can hamper the emergence of other species in the mixture, exerting, in some cases, a competition for light [66]. No significant correlation (Pearson) between thousand-seed weight and the other studied traits was observed (data not shown). This is in agreement with the work of Tudela-Isanta et al. [20] on seed performances of 53 species growing in different alpine habitats.

To summarize, among the species with germinated seeds, four to nine presented moderate to higher performances in all the studied traits, namely *C. intybus, T. officinale, L. vulgare,* and *D. carthusianorum*. Apart for germination period, *B. perennis* was also of interest, especially for higher germination rate. Moreover, even if with slower aptitude to germinate, *L. angustifolia* showed moderate values for final germination percentage and germination potential. On the opposite, *T. alpinum* had very low FGP and GP but with faster germination. Allium schoenoprasum and *D. pavonius* presented scarce performances for all the parameters. These findings indicate that a seed mixture composed by *C. intybus, T. officinale, L. vulgare, D. carthusianorum, B. perennis,* and *L. angustifolia* could be effective for the composition of seed mixtures for urban areas such as urban parks, roundabouts, and green roofs but also in buffer stripes and walls for urban vertical gardens in sub-alpine urban areas. Conversely, at higher altitude, a seed mixture containing *D. carthusianorum,*

*T. alpinum*, and *L. angustifolia* could be suggested. Due to the renewed interest in edible plants in landscape design [67], the use of these species can provide a unique ornamental component with additional health and aesthetic benefits.

## 4. Conclusions

Information provided in this study represent an important starting point for the implementation of innovation in the landscaping sector, supporting the production and trade of native flora seeds for new local green solutions, with particular reference to hilly and mountainous areas.

Overall, since germination rate and the onset of germination are the main points of wildflowers' use, apart from mesophilous hilly-mountain grasslands species, such as *B. perennis*, *C. intybus*, *L. vulgare* and *T. officinale*, also *D. carthusianorum*, and *L. angustifolia*, with distribution areas at higher altitude, germinated well under the tested controlled growing conditions. Thus, they could represent new validated candidates for wildflower application in anthropic environments, enhancing the biodiversity of urban vegetation in alpine and sub-alpine areas.

The obtained results of the controlled germination test have to be regarded as partial, because dormancy level may significantly vary among wild plant populations and years of seed collection. The possible recalcitrant or orthodox behavior of seeds under different storage conditions will also need to be investigated. In-depth studies on the effects of climatic conditions and substrates composition on seed germination performance in open-field conditions will have to be carried out. Moreover, evaluation of different parameters such as the mixture seeds rate, the relationship among the species, the habitus of each species, morphometric traits, and eco-physiology are therefore necessary to refine and enrich data on local alpine and sub-alpine wildflower species.

**Author Contributions:** Conceptualization, V.S.; methodology, M.C., S.D. and V.S.; investigation, M.C., S.D., E.M. and P.M.C.; resources, V.S.; data curation, M.C. and S.D.; writing—original draft preparation, M.C.; writing—review and editing, M.C., S.D., E.M., P.M.C. and V.S.; supervision, V.S.; funding acquisition, V.S. All authors have read and agreed to the published version of the manuscript.

**Funding:** This research was supported by the programs Interreg V-A Francia Italia Alcotra, project n. 1139: "ANTEA—Attività innovative per lo sviluppo della filiera transfrontaliera del fiore edule" and InterregFrancia Italia Alcotra project n. 8336: "ANTES".

**Institutional Review Board Statement:** Not applicable.

**Informed Consent Statement:** Not applicable.

**Data Availability Statement:** Not applicable.

**Acknowledgments:** Authors acknowledge Michele Lonati, Simone Ravetto Enri, Sophie Ghirardi, Niccolò Leonardi, and Walter Gaino for helping the sampling of the seeds.

**Conflicts of Interest:** The authors declare no conflict of interest.

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
