# Peer review of "Germination Performances of 14 Wildflowers Screened for Shaping Urban Landscapes in Mountain Areas"

_sustainability, doi:10.3390/su14052641_

Round 1

Reviewer 1 Report

The idea of the manuscript is good, especially it will give new information related to plants that can use them in the landscape.  But the English language of the manuscript should be revised by professional in English langue. 

  1. Authors should add photos of seeds and plants of each species in supplementary data 
  2. Line 14 add letter “S” to area to become (areas)
  3. Line 37 add letter “S” to year to become (years)
  4. Line 52 and line 84 add article “the” front of presence to become (the presence )
  5. Line 54 change word “surviving” to “survive”
  6. Line 55 add article “a” front of word deeper to become (a deeper)
  7. Line 55 change article “on” to “of” to become  “investigation of”
  8. Line 56 change are to is to become  (is required)
  9. Line 62 change word “landscaping” to “landscaping”
  10. Line 70 remove article “an” from an economic
  11. Line 76 remove the article “the ” front of “the vitality”
  12. Line 77 change article “for” to “in” to become (crucial role in)
  13. Line 83 add article “S” to words “area and source” to become (areas and sources)
  14. Line 85 add article “the” front of western to become (the western)
  15. Line 97 change “in” to “into” to become (stored into)
  16. Line 104 add article “S” to words “repetition” to become (repetitions)
  17. Line 106 change article “by” to “in”
  18. Line 109 change “On the basis of” to “based on”
  19. Line 128 change “in” to “into”
  20. Line 192 add article the front word  “light” to become (if the light is indicated)
  21. Line 238 add comma after word the study to become (, in the present study,)
  22. Line 280 change article “for” to “in”
  23. Line 309 correct the word “performaces” to “performances”
  24. Line 311 correct the word “Morevoer” to “Moreover”
  25. Line 332 remove the article a front of word “distribution”
  26. Line 333 remove the article a front of word “new”
  27. Line 333 change the word validate to validated
  28. Line 335 add letter “S” to area to become (areas)

Author Response

February 9th, 2022

To:

Editorial Office

Sustainability

Dear Editorial Office,

We submit a revised version of the article “Germination Performances of 14 Wildflowers Screened for Shaping Urban Landscapes in Mountain Areas” by Matteo Caser, Sonia Demasi, Eric Mozzanini, Paola Maria Chiavazza and Valentina Scariot for publication in the Special Issue “Biodiversity 2021: Agriculture, Environment and Wellbeing” of the journal “Sustainability”.

We thank for the comments and suggestions that were very helpful to further improve clarity of the manuscript.

For the preparation of the revised manuscript, we followed all the comments and suggestions of the editor and the reviewers as stated below. We highlighted the main changes to the text by red.

Reviewer 1

Point 1: The idea of the manuscript is good, especially it will give new information related to plants that can use them in the landscape.  But the English language of the manuscript should be revised by professional in English langue.

Point 1: we thank the Reviewer. The writing in the entire manuscript was checked by professional in English langue.

Point 2: Authors should add photos of seeds and plants of each species in supplementary data

Point 2: we added a new Figure 1 with the detail of each tested species in Material and Methods Section.

Point 3: Line 14 add letter “S” to area to become (areas)

Line 37 add letter “S” to year to become (years)

Line 52 and line 84 add article “the” front of presence to become (the presence )

Line 54 change word “surviving” to “survive”

Line 55 add article “a” front of word deeper to become (a deeper)

Line 55 change article “on” to “of” to become  “investigation of”

Line 56 change are to is to become  (is required)

Line 62 change word “landscaping” to “landscaping”

Line 70 remove article “an” from an economic

Line 76 remove the article “the ” front of “the vitality”

Line 77 change article “for” to “in” to become (crucial role in)

Line 83 add article “S” to words “area and source” to become (areas and sources)

Line 85 add article “the” front of western to become (the western)

Line 97 change “in” to “into” to become (stored into)

Line 104 add article “S” to words “repetition” to become (repetitions)

Line 106 change article “by” to “in”

Line 109 change “On the basis of” to “based on”

Line 128 change “in” to “into”

Line 192 add article the front word  “light” to become (if the light is indicated)

Line 238 add comma after word the study to become (, in the present study,)

Line 280 change article “for” to “in”

Line 309 correct the word “performaces” to “performances”

Line 311 correct the word “Morevoer” to “Moreover”

Line 332 remove the article a front of word “distribution”

Line 333 remove the article a front of word “new”

Line 333 change the word validate to validated

Line 335 add letter “S” to area to become (areas)

Point 3: we modified the text, accordingly.

We remain available to clarify any issue or answer that Reviewers or Editors may raise.

Best regards,

Matteo Caser,

Sonia Demasi,

Eric Mozzanini,

Paola Maria Chiavazza and

Valentina Scariot

Department of Agricultural, Forest and Food Sciences

University of Turin

Largo Paolo Braccini, 2

10095, Grugliasco (TO)

Italy

Phone number: +039-011/6708935

Fax number: +039-011/6708798

e-mail: matteo.caser@unito.it

Reviewer 2 Report

The work deals with a very interesting issue related to increasing the biodiversity of urban areas by introducing flowering plants growing wild in the mountains. The authors set out to investigate the effects of seed germination of 14 plant species in the presence or absence of light. There is not enough information on this subject in the available literature, so it is advisable and advisable to take up this issue. The results, their analysis and conclusions do not raise any objections. Another advantage of the study is the presentation of guidelines for further research in this field. However, I propose to think again about the title of the manuscript, as at present there are two little related sentences with each other. It would be good to replace them with one phrase. I propose to bring the issue of germination to the fore (because it is mainly exposed in the work), adding that it was done in terms of shaping the urban landscape in mountain areas. 

I also believe that a slightly too convoluted sentence in the Abstract section between lines 18-22 should be modified. In this section, the abbreviation GDP stands for germination period, which, in my opinion, should be removed, as the abbreviations were not given for the remaining germination rates.

In my opinion, the FGP abbreviation should not appear in the Keywords section, as it can be interpreted completely differently in different countries. I propose to replace it with another keyword, especially since many other germination rates were taken into account in the work, not just this one. 

The Materials and Methods section should provide more detail about how the harvested seeds were stored. It requires clarification, inter alia, whether they were pre-dried, kept loose or in a closure, what the humidity was before the germination test was performed. I also don't really understand the meaning of the data contained in table 1 in the Geographic coordinates column. It would be more advantageous to switch to latitude and longitude references. Data on the symbol and manufacturer of the scales used to determine the weight of the seeds are missing. The germination test was carried out under given conditions, but it was not stated why they were taken. I understand that there are no standards developed for this type of seed, but when adopting certain parameters of this process, some guidelines for other seed species were probably guided by the reference here to a specific source. 

The following minor issues should also be addressed during the revision process:

– in a given reference, a comma should be used to separate successive items, and when they are consecutive – give the range, e.g. [5–8],

– remove spaces between individual paragraphs of text,

– the text should be arranged in the manuscript in such a way that the entire table is on one page (both the title and the frame),

– tables – review the spacing between lines in tables in accordance with the journal's guidelines contained in the Microsoft Word template,

– table 2 – write down the last formula with the use of symbols, similarly as it was done in relation to the above-mentioned dependencies,

– line 118 – instead of the parenthesis ending there is a semicolon,

– line 126 – delete the year of the source's release,

– lines 139, 143, 169, 222 and 295 – instead of "-" there is "–",

– line 167 – no ending of the parenthesis,

– table 3 – provide the values of the standard deviation with the same accuracy as the average values,

– lines 202, 212 and 254 – remove an unnecessary parenthesis,

– Figure 1 – the option to place axis markers would be nice,

– References – standardize the DOI numbers; correct the formatting (1, 30, 34, 35, 37, 41, 45, 46, 49, 55, 60, 64, 65 and 67).

Author Response

February 9th, 2022

To:

Editorial Office

Sustainability

Dear Editorial Office,

We submit a revised version of the article “Germination Performances of 14 Wildflowers Screened for Shaping Urban Landscapes in Mountain Areas” by Matteo Caser, Sonia Demasi, Eric Mozzanini, Paola Maria Chiavazza and Valentina Scariot for publication in the Special Issue “Biodiversity 2021: Agriculture, Environment and Wellbeing” of the journal “Sustainability”.

We thank for the comments and suggestions that were very helpful to further improve clarity of the manuscript.

For the preparation of the revised manuscript, we followed all the comments and suggestions of the editor and the reviewers as stated below. We highlighted the main changes to the text by red.

Reviewer 2

Point 1: The work deals with a very interesting issue related to increasing the biodiversity of urban areas by introducing flowering plants growing wild in the mountains. The authors set out to investigate the effects of seed germination of 14 plant species in the presence or absence of light. There is not enough information on this subject in the available literature, so it is advisable and advisable to take up this issue. The results, their analysis and conclusions do not raise any objections. Another advantage of the study is the presentation of guidelines for further research in this field. However, I propose to think again about the title of the manuscript, as at present there are two little related sentences with each other. It would be good to replace them with one phrase. I propose to bring the issue of germination to the fore (because it is mainly exposed in the work), adding that it was done in terms of shaping the urban landscape in mountain areas.

Point 1: we thank the reviewer. We modified the title in “Germination Performances of 14 Wildflowers Screened for Shaping Urban Landscapes in Mountain Areas”.

Point 2: I also believe that a slightly too convoluted sentence in the Abstract section between lines 18-22 should be modified. In this section, the abbreviation GDP stands for germination period, which, in my opinion, should be removed, as the abbreviations were not given for the remaining germination rates.

Point 2: we modified the text in the Abstract section and we removed the abbreviation GDP.

Point 3: In my opinion, the FGP abbreviation should not appear in the Keywords section, as it can be interpreted completely differently in different countries. I propose to replace it with another keyword, especially since many other germination rates were taken into account in the work, not just this one.

Point 3: we removed FGP abbreviation and we added new keywords: alpine plant species, final germination percentage, germination period, native flowering species and relative light germination percentage.

Point 4: The Materials and Methods section should provide more detail about how the harvested seeds were stored. It requires clarification, inter alia, whether they were pre-dried, kept loose or in a closure, what the humidity was before the germination test was performed.

Point 4: we added more detailed information about the seed conservation in the paragraph “2.1 Seed collection”.

Point 5: I also don't really understand the meaning of the data contained in table 1 in the Geographic coordinates column. It would be more advantageous to switch to latitude and longitude references.

Point 5: we added latitude and longitude references in the Table 1.

Point 6: Data on the symbol and manufacturer of the scales used to determine the weight of the seeds are missing.

Point 6: we added detailed information about the used scale in the paragraph “2.2. Thousand Seed Weight (TSW)”.

Point 7: The germination test was carried out under given conditions, but it was not stated why they were taken. I understand that there are no standards developed for this type of seed, but when adopting certain parameters of this process, some guidelines for other seed species were probably guided by the reference here to a specific source.

Point 7: the guidelines used for the germination test are reported in the International Rules for Seed Testing Association (IRSTA) as indicated in the paragraph “2.3 Germination test”.

Point 8: The following minor issues should also be addressed during the revision process:

– in a given reference, a comma should be used to separate successive items, and when they are consecutive – give the range, e.g. [5–8],

– remove spaces between individual paragraphs of text,

– the text should be arranged in the manuscript in such a way that the entire table is on one page (both the title and the frame),

– tables – review the spacing between lines in tables in accordance with the journal's guidelines contained in the Microsoft Word template,

– table 2 – write down the last formula with the use of symbols, similarly as it was done in relation to the above-mentioned dependencies,

– line 118 – instead of the parenthesis ending there is a semicolon,

– line 126 – delete the year of the source's release,

– lines 139, 143, 169, 222 and 295 – instead of "-" there is "–",

– line 167 – no ending of the parenthesis,

– table 3 – provide the values of the standard deviation with the same accuracy as the average values,

– lines 202, 212 and 254 – remove an unnecessary parenthesis,

– Figure 1 – the option to place axis markers would be nice,

– References – standardize the DOI numbers; correct the formatting (1, 30, 34, 35, 37, 41, 45, 46, 49, 55, 60, 64, 65 and 67).

Point 8: we modified the text, accordingly.

We remain available to clarify any issue or answer that Reviewers or Editors may raise.

Best regards,

Matteo Caser,

Sonia Demasi,

Eric Mozzanini,

Paola Maria Chiavazza and

Valentina Scariot

Department of Agricultural, Forest and Food Sciences

University of Turin

Largo Paolo Braccini, 2

10095, Grugliasco (TO)

Italy

Phone number: +039-011/6708935

Fax number: +039-011/6708798

e-mail: matteo.caser@unito.it

Reviewer 3 Report

The manuscript presents the results of seed germination behavior of 14 flower species of Alpine and Subalpine Areas, evaluating the effect of light on the main germination parameters. The results contribute to the planning of urban projects using native flora.

In general terms, the document is well presented and contains a robust experimental design that strengthens the significance of the results obtained.

One of the limitations of this work is the lack of additional studies of those cases where some dormancy behavior is presumed, however, given this possibility, the authors mention that this competence is part of a later study of greater detail in the selected species.

Additionally, it would be convenient to make some mention about the possible recalcitrant behavior of the species, given that the collection of the seeds was carried out in 2017, the samples were kept at 4oC and the germination studies lasted until 2018.

Author Response

February 9th, 2022

To:

Editorial Office

Sustainability

Dear Editorial Office,

We submit a revised version of the article “Germination Performances of 14 Wildflowers Screened for Shaping Urban Landscapes in Mountain Areas” by Matteo Caser, Sonia Demasi, Eric Mozzanini, Paola Maria Chiavazza and Valentina Scariot for publication in the Special Issue “Biodiversity 2021: Agriculture, Environment and Wellbeing” of the journal “Sustainability”.

We thank for the comments and suggestions that were very helpful to further improve clarity of the manuscript.

For the preparation of the revised manuscript, we followed all the comments and suggestions of the editor and the reviewers as stated below. We highlighted the main changes to the text by red.

Reviewer 3

Point 1: The manuscript presents the results of seed germination behavior of 14 flower species of Alpine and Subalpine Areas, evaluating the effect of light on the main germination parameters. The results contribute to the planning of urban projects using native flora.

In general terms, the document is well presented and contains a robust experimental design that strengthens the significance of the results obtained.

One of the limitations of this work is the lack of additional studies of those cases where some dormancy behavior is presumed, however, given this possibility, the authors mention that this competence is part of a later study of greater detail in the selected species.

Additionally, it would be convenient to make some mention about the possible recalcitrant behavior of the species, given that the collection of the seeds was carried out in 2017, the samples were kept at 4° C and the germination studies lasted until 2018.

Point 1: we thank the reviewer. In the Conclusion Section we mentioned that a thorough study on the recalcitrant or orthodox behavior of these seeds at different conservation conditions will have to be conducted in the future.

We remain available to clarify any issue or answer that Reviewers or Editors may raise.

Best regards,

Matteo Caser,

Sonia Demasi,

Eric Mozzanini,

Paola Maria Chiavazza and

Valentina Scariot

Department of Agricultural, Forest and Food Sciences

University of Turin

Largo Paolo Braccini, 2

10095, Grugliasco (TO)

Italy

Phone number: +039-011/6708935

Fax number: +039-011/6708798

e-mail: matteo.caser@unito.it

Round 2

Reviewer 1 Report

the authors replied well to the comments and did all the required corrections. I recommend accepting the manuscript for publication